behaviour, ecology, evolution

relative rank, proportional rank, standardized rank, longitudinal studies, social dominance, baboons

**Author for correspondence:**
Elizabeth A. Archie
e-mail: earchie@nd.edu

†The first two authors contributed equally to this work.

‡The last two authors contributed equally to this work.

# A comparison of dominance rank metrics reveals multiple competitive landscapes in an animal society

Emily J. Levy[1,†], Matthew N. Zipple[1,†], Emily McLean[3], Fernando A. Campos[1,4], Mauna Dasari[5], Arielle S. Fogel[2,6], Mathias Franz[7], Laurence R. Gesquiere[1], Jacob B. Gordon[1], Laura Grieneisen[8], Bobby Habig[5,9], David J. Jansen[5], Niki H. Learn[10], Chelsea J. Weibel[5], Jeanne Altmann[10,11], Susan C. Alberts[1,2,11,‡] and Elizabeth A. Archie[5,11,‡]

[1]Department of Biology, and [2]Department of Evolutionary Anthropology, Duke University, 130 Science Drive, Durham, NC 27708, USA
[3]Division of Natural Sciences and Mathematics, Oxford College of Emory University, 801 Emory Street, Oxford, GA 30054, USA
[4]Department of Anthropology, University of Texas at San Antonio, One UTSA Circle, San Antonio, TX 78249, USA
[5]Department of Biological Sciences, University of Notre Dame, Notre Dame, IN 46556, USA
[6]University Program in Genetics and Genomics, Duke University, 3 Genome Court, Durham, NC 27710, USA
[7]Institute for Biology, Freie Universitaet Berlin, Königin-Luise-Strasse 1-3, D-14195 Berlin, Germany
[8]College of Biological Sciences, University of Minnesota, 420 Washington Ave. SE, Minneapolis, MN 55455, USA
[9]Department of Biology, Queens College, City University of New York, 65-30 Kissena Blvd., Flushing, New York, NY 11367, USA
[10]Department of Ecology and Evolutionary Biology, Princeton University, 106A Guyot Hall, Princeton, NJ 08544, USA
[11]Institute of Primate Research, National Museums of Kenya, Nairobi 00502, Kenya

EJL, 0000-0002-8182-9456; MNZ, 0000-0003-3451-2103; FAC, 0000-0001-9826-751X; MD, 0000-0002-1956-2500; ASF, 0000-0003-3048-7959; JBG, 0000-0001-6963-4405; LG, 0000-0001-7286-5001; BH, 0000-0003-0486-4482; DJJ, 0000-0001-5857-0581; CJW, 0000-0002-1089-3215; SCA, 0000-0002-1313-488X; EAA, 0000-0002-1187-0998

Across group-living animals, linear dominance hierarchies lead to disparities in access to resources, health outcomes and reproductive performance. Studies of how dominance rank predicts these traits typically employ one of several dominance rank metrics without examining the assumptions each metric makes about its underlying competitive processes. Here, we compare the ability of two dominance rank metrics—simple ordinal rank and proportional or 'standardized' rank—to predict 20 traits in a wild baboon population in Amboseli, Kenya. We propose that simple ordinal rank best predicts traits when competition is density-dependent, whereas proportional rank best predicts traits when competition is density-independent. We found that for 75% of traits (15/20), one rank metric performed better than the other. Strikingly, all male traits were best predicted by simple ordinal rank, whereas female traits were evenly split between proportional and simple ordinal rank. Hence, male and female traits are shaped by different competitive processes: males are largely driven by density-dependent resource access (e.g. access to oestrous females), whereas females are shaped by both density-independent (e.g. distributed food resources) and density-dependent resource access. This method of comparing how different rank metrics predict traits can be used to distinguish between different competitive processes operating in animal societies.

## 1. Introduction

In group-living animals, individuals can often be linearly ranked according to their priority of access to resources or their ability to win conflicts (e.g. insects [1,2], crustaceans [3,4], fish [5,6], birds [7,8] and mammals [9,10]). The resulting

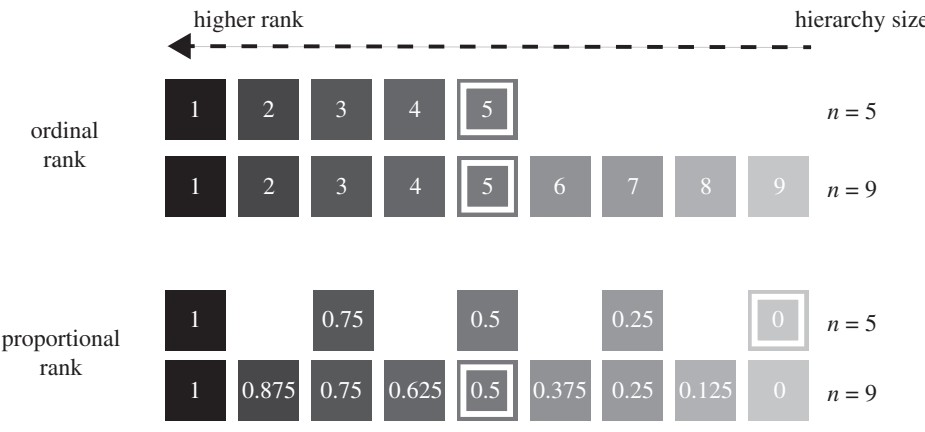

**Figure 1.** Differences between proportional and simple ordinal rank in two differently sized hierarchies. Ranks with darker shading have a competitive advantage over those with lighter shading. The fifth-ranking individual in each hierarchy is demarcated with a white border. Under a simple ordinal rank framework, being ranked fifth confers the same competitive advantages independent of hierarchy size. Under a proportional rank framework, being ranked fifth is more advantageous in a hierarchy of 9 (proportional rank = 0.5) than in a hierarchy of 5 (proportional rank = 0). Adapted from Levy et al. [33].

dominance hierarchies are associated with a wide range of traits, including physiology [11,12], immunity and disease risk [13,14], behaviour [15,16], reproductive success [9,15], longevity [15,17] and offspring survival [15,18,19]. The causes and consequences of dominance rank are therefore integral to our understanding of the evolution of animal behaviours and life-history strategies.

When studying these causes and consequences, particularly in societies with linear hierarchies, researchers commonly use any one of several ordinal or cardinal rank metrics. Ordinal rank metrics designate each individual's order in the dominance hierarchy, either by using the simple integer order (1 through $n$, hereafter called 'simple ordinal rank') or by scaling the integer order by group size (producing 'proportional' rank, also referred to as 'relative' or 'standardized' rank; e.g. [20–28]). Cardinal rank metrics employ a range of possible approaches that allow researchers to estimate not only an ordinal ranking, but also the magnitude of power differences between adjacently ranked individuals [29–31].

Often, researchers choose one of these dominance rank metrics without stating the assumptions that the metric makes about the nature of rank-based competition [20–23,25] (but see [26,32,33]). The choice of a given rank metric is important because studies sometimes find differences in the ability of different rank metrics to predict rank-related traits, even in the same population. For example, Archie et al. [26] demonstrated that proportional rank, but not simple ordinal rank, predicted risk of injury in female baboons in the Amboseli ecosystem in Kenya [26]. In the same population, proportional rank was also a better predictor of females' faecal glucocorticoid concentrations than simple ordinal rank [33]. These studies highlight the need to understand the contexts in which one rank metric predicts a trait better than another.

Here, we examine the ability of two different rank metrics to predict 20 sex- and age-class-specific traits in the Amboseli baboon population (electronic supplementary material, table S1). To keep the scope of our analysis reasonable, we focus on two ordinal rank metrics rather than cardinal metrics, but we discuss potential extensions to cardinal metrics in the Discussion section. Specifically, we compare simple ordinal rank (an integer-order system, commonly referred to as just 'ordinal rank') with the metric most commonly known as 'relative' or 'standardized' rank, but which we refer to as 'proportional' rank because this term describes more

precisely the nature of the metric. We had two goals. First, we explicitly identify the assumptions each metric makes about the underlying competitive landscapes that shape rank-related traits. In doing so, we identify theoretical scenarios in which we expect either simple ordinal or proportional rank to be a better measure of competitive interactions and, therefore, a better predictor of rank-related traits. Second, we identify which rank metric (simple ordinal or proportional) best predicts a wide range of rank-related traits in wild baboons, with the aim of better understanding density-dependent versus density-independent patterns of resource distribution and access in a complex animal society.

## (a) Assumptions of simple ordinal rank and proportional rank

As described above, an individual's simple ordinal rank reflects the *order* in which an individual appears in a linear dominance hierarchy (i.e. ranks 1, 2, 3 … $n$, where $n$ is the total number of individuals in the hierarchy; figure 1) [8,34,35]. By contrast, proportional rank accounts for the number of individuals being ranked (i.e. it accounts for hierarchy size) by measuring the *proportion* of other individuals in a hierarchy that an individual outranks (figure 1) [20–28]. For example, an individual with proportional rank 0.75 outranks 75% of other individuals in its hierarchy. When the number of individuals in the hierarchy does not vary in a given dataset, simple ordinal and proportional ranks are perfectly correlated. However, if the study contains multiple social groups with different hierarchy sizes, or if hierarchy size varies over time, then simple ordinal and proportional ranks are no longer interchangeable (see electronic supplementary material 'Identifying changes in the relationship between simple ordinal and proportional ranks over time' and figure S4).

As a theoretical example of a situation in which simple ordinal and proportional ranks are not interchangeable, consider a hierarchy that contains five males. Those males will have simple ordinal ranks 1–5 and proportional ranks 1, 0.75, 0.5, 0.25 and 0 (figure 1, $n = 5$). If, over time, four more males join the group and are ranked at the bottom of the hierarchy, the simple ordinal ranks of the original five males will remain the same, but their proportional ranks in the larger hierarchy will be 1, 0.875, 0.75, 0.625 and 0.5 (figure 1, $n = 9$;

Proc. R. Soc. B 287: 20201013

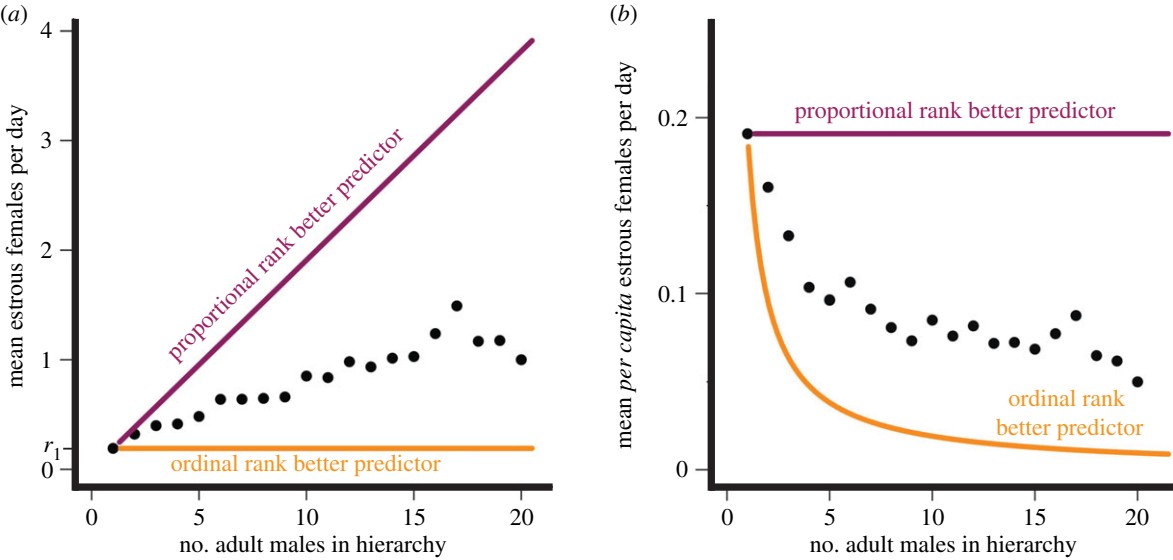

**Figure 2.** (a) The theoretical and empirical relationships between male hierarchy size (x-axis) and resource availability (y-axis) using the example of oestrous female baboons, a resource over which male baboons compete for mating success. The orange line shows a theoretical scenario in which the number of oestrous females in the group (total resource base) is constant as the number of males in the hierarchy increases; in this case, male mating success (the resulting measured trait) would be predicted by simple ordinal rank. The purple line shows a scenario in which the number of oestrous females increases in proportion to the number of males in the hierarchy; in this case, male mating success would be predicted by proportional rank. The slope of the orange line is 0 and the intercept is $r_1$, which designates the quantity of resources available in a hierarchy size of 1 male ($r_1 = 0.2$ oestrous females in this figure). This value, $r_1$, determines the slope of the purple line; i.e. for proportional rank to perfectly predict mating success, resource availability must increase by $r_1$, the quantity available to the first male, as each male is added to the hierarchy. The empirical relationship between male hierarchy size and the number of oestrous females (Amboseli baboon data; black points) is positive, but the slope is closer to the orange line than the purple line. Thus, we expect simple ordinal rank to best predict mating success. (b) Similar to (a), but the number of oestrous females is plotted per capita (i.e. per adult male in the hierarchy). The orange curve illustrates the case in which the resource stays constant across different hierarchy sizes; thus, average per capita resource access declines as hierarchy size increases. The purple line illustrates the case in which the resource base increases proportionately with hierarchy size; thus, average per capita resource access is fixed. The black points represent the same empirical data as in (a). Note that the framework above assumes that any given individual's ability to maintain control of a resource is independent of group size. (Online version in colour.)

electronic supplementary material, figure S4). In this situation, a researcher who uses simple ordinal rank would conclude that the fifth-ranking male in the hierarchy remained in a constant competitive position throughout the entire study period, whereas a researcher who uses proportional rank would conclude that the fifth-ranking male transitioned from a rank of 0 to 0.5, a major change in dominance rank. Which researcher is correct? The answer depends on the nature of the competitive interactions for which dominance rank serves as a proxy.

The relationship between hierarchy size and resource availability is integral to the assumptions underlying the use of simple ordinal versus proportional rank metrics (figure 2; electronic supplementary material, table S5). Using simple ordinal rank assumes that the resource base over which individuals compete will not increase as group size increases (figure 2, orange lines). The result will be more intense competition, on average, in larger groups and a worse outcome for the lowest-ranking individuals in larger compared with smaller groups. In this scenario, the most salient dominance measure for a focal individual is how many individuals are ranked above that individual. For example, in non-synchronous, non-seasonal breeders such as baboons, no more than one or two females are likely to be in oestrus on any given day, even in groups with many females (other females may be pregnant, lactating or in a non-oestrous phase of their cycle). As group size increases, the daily availability of oestrous females increases more slowly than either the number of adult females or the number of adult males in the group (figure 2a). The result is a decline in males' average per capita access to oestrous females as male hierarchy size

increases (figure 2b). If the male dominance hierarchy functions like a queue in which males wait for mating opportunities, a male's mating opportunities will not depend on the number of other males in his hierarchy per se, but instead upon the number of males that are ranked above him [36] (electronic supplementary material, table S5, competition for mates). In other words, the fifth-ranking male in a hierarchy of 5 will have the same mating access as the fifth-ranking male in a hierarchy of 9. When average per capita resource access is density-dependent, we expect simple ordinal rank to be a better measure of competition and a better predictor of traits determined by that competition compared with proportional rank.

By contrast, when average per capita resource availability is density-independent, such that a larger hierarchy has a proportionately larger resource base, we expect proportional rank to be a better measure of competition and a better predictor of traits determined by that competition compared with simple ordinal rank (figure 2, purple lines). This situation might occur, for instance, in competition for food if a hierarchy grows from five to nine individuals and its home range nearly doubles in size (with nearly twice the amount of food). Unlike our mate competition example, the third-ranking individual in the hierarchy of 5 has approximately equal access to food as the fifth-ranking individual in the group of 9. In this scenario, the most salient dominance measure for a focal individual is the *proportion* of individuals that it outranks. The individual ranked 5 of 9 is outranked by four individuals, and the individual ranked 3 of 5 is outranked by only two individuals, but both are dominated by 50% of their group mates, and the greater resource base of

the larger group means that these two individuals experience approximately the same resource access (figure 2, purple lines; electronic supplementary material, table S5, competition for food).

Density-dependent competition occurs when average per capita resource access depends on hierarchy size (i.e. when resources do not increase proportionately with increases in hierarchy size), while density-independent competition occurs when average per capita resource access is independent of hierarchy size (because resources increase proportionately with hierarchy size). We therefore predict that some rank-related traits will be better predicted by simple ordinal rank and others will be better predicted by proportional rank. Furthermore, this difference in predictive power should reflect the underlying competitive processes that shape the resulting traits—specifically, the relationship between hierarchy size and resource base. We assess this prediction by examining 20 traits measured as part of a long-term longitudinal study of a wild baboon population, in which both sexes form linear dominance hierarchies. We make two specific predictions about when we expect simple ordinal versus proportional rank to best predict a given set of traits. First, because the average per capita number of oestrous females in a group does not increase proportionately with male hierarchy size, we predict that male traits associated with mate competition should be better predicted by simple ordinal rank than by proportional rank (figure 2) [37]. Second, we predict that female traits should be better predicted by proportional rank than by simple ordinal rank. We make this prediction because home range size (and thus access to food) increases roughly in proportion with group size [38], and because socioecological models predict that food competition is the most salient form of intra-sex competition for females [39], although other factors are also important (e.g. [40]).

To date, only a handful of studies have tested the ability of different rank metrics to predict traits of interest [26,32,33,41–43]. Several other studies have used simulated or empirical data to assess whether different rank metrics produce different hierarchies, given the same data (e.g. [44–48]). By comparing differences in the ability of simple ordinal and proportional rank metrics to predict 20 traits in a population of wild baboons, we perform the most extensive comparison to date of the ability of different rank metrics to predict traits. Our findings reveal a new way to understand competition acting in a gregarious animal society and facilitate the generation of new hypotheses about the way that competition shapes a diverse range of traits in social species.

## 2. Methods

### (a) Study population

The Amboseli Baboon Research Project is a long-term study of a natural population of savannah baboons located in Kenya's Amboseli basin. Data collection began in 1971 and continues today [49]. The population consists primarily of yellow baboons (*Papio cynocephalus*) that experience some naturally occurring admixture with olive baboons (*P. anubis*) [50–52]. The number of social groups under observation at any given time has ranged from 1 to 6, varying as a result of logistical considerations or group fissions and fusions. All individuals in study groups are visually recognized based on morphological and facial features. Near-daily demographic, environmental and behavioural data have been collected throughout the study, and paternity data

(beginning around 1995) and endocrinological data (beginning around 2000) have been collected for part of the study.

### (b) Calculation of dominance rank

We routinely calculate both simple ordinal ranks and proportional ranks for males and females on a monthly basis. Only adult ranks are considered in this analysis. For traits measured in immature individuals, maternal dominance rank is used as the predictor variable.

Dominance ranks are determined by assigning wins and losses in dyadic agonistic interactions between same-sex individuals. Data on agonistic interactions are collected ad libitum during daily data collection, typically while the observer is simultaneously carrying out random-order focal animal sampling [53]. This sampling procedure ensures that observers continually move to new locations within the social group and observe focal individuals on a regular rotating basis. An individual is considered to win an agonistic interaction if they displace another individual, or if they give only aggressive or neutral gestures while their opponent gives only submissive gestures. All agonistic outcomes are entered into sex-specific dominance matrices (i.e. adult males are ranked separately from adult females). Individuals are placed in order of descending, sex-specific rank so as to minimize the number of entries that fall below the diagonals of the matrices [37,54].

Simple ordinal ranks are produced by numbering individuals according to the order in which they occur in the monthly matrix (1, 2, 3 … $n$, where $n$ = hierarchy size), with 1 being the highest-ranking male or female in the hierarchy and $n$ being the lowest. Proportional ranks are computed as $1 - ((\text{simple ordinal rank} - 1)/(\text{hierarchy size} - 1))$ to produce ranks that fall in the range of [0,1] for every hierarchy, with 1 being the highest-ranking male or female in the hierarchy and 0 being the lowest.

### (c) Re-analysis of previous studies

We aimed to test whether 20 different sex- and age-class-specific traits were better predicted by simple ordinal rank or proportional rank in the Amboseli baboon population. We first identified previous publications from the Amboseli Baboon Research Project that reported statistically significant effects of rank on various traits. For a complete list of re-analyses performed, see electronic supplementary material, table S1.

Our methods of re-analysis followed three steps:

1. We replicated as closely as possible the dataset used to produce the original analyses. In the case of datasets stored on the Dryad Digital Repository (datadryad.org), these datasets could be matched exactly (see electronic supplementary material, table S1). If the original dataset was not deposited on Dryad, we re-extracted the dataset as well as we could from the Amboseli Baboon Research Project's long-term, relational database. However, the datasets we extracted were sometimes slightly different from those originally analysed because the database changes slightly over time as corrections are made. In all cases, we produced qualitatively close matches to the originally reported dataset in terms of sample sizes and summary statistics.

2. We replicated as closely as possible the statistical calculations presented in the original analysis (electronic supplementary material, table S6). All re-analyses were carried out in R [55], even though some original analyses were carried out in SPSS, JMP or SAS. To maintain consistency across all analyses reported here, all linear models, general linear models and mixed-effects models were built using the function *glmmTMB* [56]. All survival models were built using the function *coxph* [57]. In some cases, differences between the original study and our replication, either because of software differences or dataset

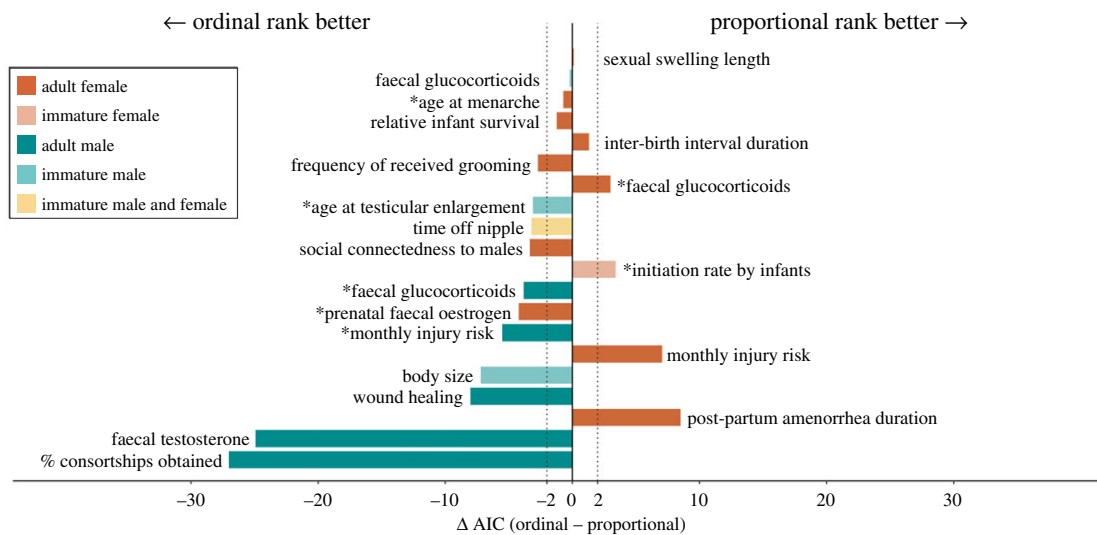

**Figure 3.** Visualization of model outcomes when predicting the same trait with simple ordinal rank (labelled 'ordinal rank') versus proportional rank. Each bar corresponds to a trait, and its value corresponds to a difference in AIC scores between models that used simple ordinal versus proportional rank. Vertical dashed lines represent |ΔAIC| = 2. For traits whose bars are within the dashed lines, neither rank metric performed substantially better than the other (5/20 analyses; we did not find any indication that the ability of the models to differentiate between the predictive power of simple ordinal versus proportional rank depended on the duration of the study; $p = 0.9$, Pearson's product moment correlation). For traits whose bars are to the left of the dashed lines, simple ordinal rank was a better predictor of the trait than proportional rank (11/20), and vice versa for traits whose bars are to the right of the dashed lines (4/20). Colours of bars indicate sex (male, female and both), and shading indicates age class (adult or maternal rank of immatures). Asterisks indicate the seven traits for which only one rank metric predicted the trait better than the null model. The top two bars, sexual swelling length and faecal glucocorticoids, were traits measured in adult females and immature males, respectively. See electronic supplementary material, table S1 for information about the originally identified rank effects. (Online version in colour.)

differences, caused our replicated models to be slightly different from the original models. However, our re-analyses were qualitatively consistent with the original analyses.

3. For each of the models described in step 2, we built two additional alternative models: (i) a model that replaced the rank term used in the original model with the alternative rank metric (proportional rank if simple ordinal rank was originally used and vice versa). (ii) A null model that removed the rank term from the model. We then extracted AIC values from all three models to determine which model, if any, best fitted the data. We interpreted an AIC difference of ≥2 to mean that one model was preferred over another, with preference for the model with a lower AIC score. This 2-unit cut-off is standard practice and approximates a $p$-value of 0.05 [58,59].

## 3. Results

### (a) Rank metrics differ in their ability to predict traits

Nineteen of the 20 traits were better predicted by one or both rank metrics than by the null model (ΔAIC$_{Preferred – Null}$ ≤ −2); for one trait (adult female faecal glucocorticoids), ΔAIC between the preferred metric and the null model was −1.4 (electronic supplementary material, table S1) [33]. For 15 of the 20 traits (75%), we found that one of the two rank metrics—simple ordinal or proportional—performed substantially better than the other in predicting a given trait (|ΔAIC| ≥ 2; figure 3; electronic supplementary material, table S1). In addition, in 7 of these 15 models, only one of the two rank metrics performed better than the null model. This means that for 35% of traits (7 of 20), a relationship between rank and a trait of interest would have been undetected if researchers had chosen the alternative rank metric. For example, male faecal glucocorticoid concentrations were predicted by simple ordinal rank (ΔAIC$_{Simple\ ordinal – Null}$ = −3.6), but not by proportional rank (ΔAIC$_{Proportional – Null}$ = 0.2).

### (b) All male traits are better predicted by simple ordinal rank

Whether proportional or simple ordinal rank was a better predictor of a trait depended on the sex of the study individuals, supporting our predictions that male and female baboons experience different competitive regimes. Of the seven male traits that were better predicted by one rank metric than the other, all 7 (100%) were best predicted by simple ordinal rank (male versus chance, $p = 0.02$, two-tailed binomial test). By contrast, of the seven female traits that were better predicted by one rank metric than the other, 4 (57%) were best predicted by proportional rank, and 3 (43%) were best predicted by simple ordinal rank (male versus female, $p = 0.07$, Fisher's exact test; female versus chance, $p = 1.00$, two-tailed binomial test). In two of the three cases where traits could be directly compared between adult males and females (faecal glucocorticoid concentrations and monthly injury risk), male traits were better predicted by simple ordinal rank, whereas female traits were better predicted by proportional rank. Additionally, the two traits with the largest AIC difference between rank metrics were the percentage of consortships obtained by males and male faecal testosterone; both of these traits were best predicted by simple ordinal rank (ΔAIC = 27 and 25, respectively; electronic supplementary material, table S1).

### (c) All traits related to social and/or mating partners are best predicted by simple ordinal rank

A second pattern that emerged from these results is that competition for social and mating partners in both sexes was better predicted by simple ordinal rank than by proportional rank. Specifically, simple ordinal rank was a better predictor for all three traits that can be interpreted in terms of access to

social and mating partners—male percentage of consortships obtained, female social connectedness to males and female frequency of received grooming from males or females.

## 4. Discussion

Simple ordinal and proportional rank metrics make different assumptions about competitive regimes in animal societies. When average per capita resource access is density-dependent, simple ordinal rank should predict competition-related traits. By contrast, when average per capita resource access is density-independent, proportional rank should predict competition-related traits. In reality, competition within animal social groups, which experience dynamic, ongoing changes in group size and resources, will rarely be purely density-dependent or density-independent. Instead, most competition will reflect a mixture of these two regimes. This point is illustrated in figure 2 for one resource important to males (number of oestrous females); neither density-dependence nor density-independence perfectly describes the relationship between group size and resource availability. Nonetheless, in many contexts, one or the other competitive regime will predominate. In support, we have shown that proportional and simple ordinal rank metrics differ in how well they predict 75% (15/20) of rank-related traits examined in the Amboseli baboon population. Strikingly, in 35% of examined traits (7/20), only one of the two rank metrics was predictive of the trait, meaning that researchers could have failed to identify a rank-related effect if they considered only a single metric. In addition, our data indicate that male and female traits are often shaped by different competitive regimes. Below, we discuss these sex differences in more detail.

### (a) Sex differences in competitive regimes in baboons

Males' competitive environments appear to be frequently shaped by density-dependent resource access, as evidenced by the strong and consistent performance of the simple ordinal rank metric in predicting many male phenotypes. This finding supports our first prediction that traits shaped by competition for oestrous females should be better modelled with simple ordinal rank. Indeed, of the 20 traits we measured, male mate guarding success (i.e. percentage of consortships obtained) and faecal testosterone concentrations are most directly associated with male competition for mates [60,61], and these two traits had the greatest difference in AIC favouring simple ordinal rank ($\Delta$AIC for the percentage of consortships = 27; $\Delta$AIC for faecal testosterone = 25). Several other male traits are indirectly associated with mate competition and were also better predicted by simple ordinal rank, including age at testicular enlargement, male faecal glucocorticoid concentration and male monthly injury risk [61–63]. The ability of male baboons to obtain consortships with females approximates a queuing system [37], such that a highly salient variable that affects a male's mating success is the number of males that rank higher than him. This type of competitive environment is consistent with our understanding of the contexts in which simple ordinal rank will be a better predictor of resource availability than proportional rank (see 'Assumptions of simple ordinal rank and proportional rank' above; figure 2).

In partial support of our second prediction, we found that female competitive environments are often shaped by density-independent competition: in over half the female traits in

which one of the two rank metrics performed better, proportional rank received stronger support than simple ordinal rank. Although we do not have a direct measure of food competition, three of the traits that were better predicted by proportional rank are likely to be associated with food competition, namely adult female faecal glucocorticoid concentrations (which can indicate food stress [33]), monthly injury risk among adult females (which is indicative of conflict over resources [26]) and the duration of post-partum amenorrhoea (which is tightly linked to energy balance [64]). However, four female traits were predicted equally well by both rank metrics, and three were better predicted by simple ordinal rank (frequency of received grooming, social connectedness to adult males and prenatal faecal oestrogen). Thus, only some female traits reflect density-independent competitive processes; other female traits are likely to be shaped by density-dependent competition, or are the result of both density-dependent and density-independent competition, perhaps indicating that females are competing for resources other than food.

Related to this point, our results suggest that, for both sexes, average per capita access to social and mating partners decreases as hierarchy size increases; that is, competition for social and mating partners may be best understood as a density-dependent process. In addition to the male traits already discussed, both of the female traits that can be interpreted in terms of access to social partners were better described by simple ordinal rank (frequency of received grooming and social connectedness to adult males). This observation highlights the potential for our approach to help generate new hypotheses and predictions that can be tested to understand the key factors (e.g. access to food and mates) that shape within-group competition. For example, we hypothesize that traits related to social interactions often reflect female competition for social partners, and that 'prime' social partners are a limiting resource in a social group, such that the number of prime social partners does not increase in proportion with group size.

### (b) Implications and potential extensions of our study

Because proportional and simple ordinal ranks reflect different assumptions about the competitive processes influencing social animals, the methods we use here can be applied in other social systems to inform researchers' understanding of the competitive processes operating in their study species. A researcher who compares proportional and simple ordinal rank models and finds that simple ordinal rank is a much stronger predictor of a trait (e.g. male access to females; figures 2 and 3; electronic supplementary material, table S5) can conclude that average per capita access to the resource declines as hierarchy size increases, and that competition for that resource is primarily a density-dependent process. By contrast, a finding that proportional rank better explains a trait (e.g. post-partum amenorrhoea duration in females; figure 3) allows a researcher to conclude that the trait is shaped primarily by density-independent competitive processes, such that per capita access to resources are relatively constant across hierarchy sizes. These methods and logic can also be applied to other rank metrics, such as cardinal ranks or coding individuals as alpha or non-alpha. Each metric assumes a different underlying competitive process—for example, coding individuals as alpha (highest-ranking) or

non-alpha assumes that the alpha individual experiences a different level of resource competition than all others in the hierarchy, who in turn experience comparable resource competition with each other. Models that use each metric can then be compared via an AIC score similarly to the present study.

Our study is the first systematic comparison of the ability of different dominance rank metrics to predict numerous traits in the same population. Proportional and simple ordinal ranks have rarely been explicitly compared; to our knowledge, only five studies, all in primate species, have previously compared the predictive ability of these two rank metrics. Two studies found that proportional rank better predicted the phenotypes in question than did simple ordinal rank (male consortship rates in rhesus macaques [32] and rates of injury among female baboons [26]). A third study, in female baboons, found that proportional rank better predicted faecal glucocorticoid concentrations than did simple ordinal rank, but whether a female had alpha status or not was an even better predictor than proportional rank [33]. Similarly, a fourth study reported that a 'high versus low' categorical measure of rank better predicted female feeding time in rhesus macaques than did proportional or simple ordinal rank, with high-ranking females spending more time feeding than low-ranking females [41]. A fifth study found that neither proportional nor simple ordinal rank was a statistically significant predictor of the probability of conception in female blue monkeys [42]. In addition, several method-based studies have tested whether rank orders differ depending on which metric is used to calculate dominance rank, but these have not used empirical data to compare how rank metrics perform in predicting traits (e.g. [44–48]; but see [43]).

Our results also point to the value of long-term, individual-based research [65,66]. Without many years of data or data from multiple social groups, we would have been unable to detect differences in the explanatory power of proportional versus simple ordinal rank metrics. Through the comparison of these two metrics, we are able to gain a deeper understanding of the sex-specific competitive environments shaping different traits in our study population. We see the previously unappreciated differences in proportional and simple ordinal rank metrics not as a weakness of research that has already been performed, but as a new tool that can be employed in the study of diverse systems.

Our findings also have implications for meta-analyses and comparative studies of rank-related effects (e.g. [15,16,67]). It is paramount that, before including studies that employ different measures of rank, a meta-analyst considers whether rank metrics presented across multiple studies are equivalent. For example, studies that report effects of rank for 'high'- versus 'low'-ranking individuals create category thresholds based on either proportional or simple ordinal ranks, depending on whether 'high' and 'low' refers to social position relative to the whole population (simple ordinal rank) or to each social group individually (proportional rank). Furthermore, if a study is reporting on only a single social group over a short time period, then hierarchy size is likely to be constant and therefore simple ordinal and proportional ranks would be equivalent. However, if a study is reporting on multiple study groups or even a single study group over a long time period, then rank metrics may no longer be interchangeable. We therefore recommend that meta-analysts assembling datasets from multiple studies should (i) carefully consider the underlying assumptions that link rank metrics to competitive

landscapes in order to determine which rank metric is most appropriate, and (ii) include only studies with equivalent rank metrics in a given meta-analysis, converting between rank metrics when possible and necessary. When following these recommendations is impracticable, meta-analysts should acknowledge the limitations of drawing inferences from studies with non-equivalent rank metrics.

Finally, our theoretical framework and analyses only consider two ordered metrics of social rank; we do not consider cardinal rank metrics, such as Elo rating or David's score, which do not assume equal distance in the hierarchy between consecutively ranked individuals [29–31]. While a detailed examination of the latent assumptions in these metrics and a comparison of their ability to predict rank-related phenotypes are beyond the scope of this paper, our results may also have implications for the selection of cardinal rank metrics. For example, a researcher that employs Elo ratings as their measure of dominance rank must decide whether to use an animal's raw Elo rating or a standardized Elo rating (i.e. scale between 0 and 1); the standardized score accounts for differences in group size or differences in the within-group range of Elo rating (see [48] for some discussion). Future work should seek to determine how the choice of standardized versus absolute cardinal rank metrics fits into the theoretical framework we outline here (e.g. how cardinal ranks map on to figure 2).

We hope that our findings encourage other researchers working on long-term studies to perform similar analyses comparing the predictive power of proportional and simple ordinal rank metrics. We also encourage researchers to consider and explicitly state the latent assumptions that are made by using any particular rank metric and to consider if their traits of study are more likely to be explained by one rank metric versus another.

Ethics. Our research is approved by the Institutional Animal Care and Use Committees (IACUC) at Duke University, the University of Notre Dame and Princeton University. We adhere to all the laws and guidelines of Kenya and to the Guidelines for the Treatment of Animals in Behavioural Research and Teaching established by the Animal Behaviour Society [68].

Data accessibility. Data necessary to reproduce figures and analyses are included in electronic supplementary material, table S1.

Authors' contributions. E.J.L., M.N.Z., E.A.A. and S.C.A. created the theoretical framework. M.N.Z. conceived of the re-analysis of prior data and performed the $R^2$ analysis. M.N.Z., E.J.L. and E.M. organized the re-analysis. E.J.L., M.N.Z., E.M., F.A.C., M.D., A.S.F., L.R.G., L.G., B.H., D.J.J. and C.J.W. carried out re-analyses. M.F. provided statistical guidance. J.B.G., D.J.J. and N.H.L. provided logistical support for re-analyses. E.J.L. and M.N.Z. finalized analyses and dataset. F.A.C., E.J.L. and M.N.Z. created visualizations. J.A. and S.C.A. carried out long-term data collection to enable re-analyses. M.N.Z., E.J.L., S.C.A. and E.A.A. drafted the manuscript; all co-authors revised the manuscript and gave final approval for publication.

Competing interests. We declare we have no competing interests.

Funding. We gratefully acknowledge the support of the National Science Foundation and the National Institutes of Health for the majority of our data collection. Over the past several decades, we acknowledge the NSF support from grant nos IOS 1456832, IOS 1053461, DEB 1405308, IOS 0919200, DEB 0846286, DEB 0846532, IBN 0322781, IBN 0322613, BCS 0323553, BCS 0323596, IBN 9985910, IBN 9422013, IBN 9729586, IBN 9996135 and IBN 9985910. At the NIH, we are grateful for support from the National Institute on Aging (grant nos R01AG053330, R21AG055777, P01AG031719, R21AG049936, R03AG045459 and R01AG034513-01), the National Institute of Child Health and Development (grant no. R01HD088558) and the Princeton Center for the Demography of Aging (P30AG024361). We also thank Duke University, Princeton University, the University of Notre Dame, the Chicago Zoological Society, the Max Planck Institute for Demographic Research, the

L. S. B. Leakey Foundation and the National Geographic Society for support at various times over the years.

Acknowledgements. This project would not have been possible without long-term collaborators who have helped us maintain this project for nearly 50 years. We also thank the Kenya Wildlife Service, University of Nairobi, Kenya Institute of Primate Research, National Museums of Kenya, the Kenya National Council for Science, Technology and Innovation, and the Enduimet Wildlife Management Association. We also thank the members of the Amboseli-Longido pastoralist communities, Ker & Downey Safaris, Air Kenya and Safarilink for their cooperation and assistance in the field. Particular thanks go to the Amboseli Baboon Research Project long-term field team—R. S. Mututua, S. N. Sayialel, J. K. Warutere and I. L. Siodi—and to T. Wango and V. Oudu for their untiring assistance in Nairobi. The baboon project database, Babase, is designed and programmed by K. Pinc. This research was approved by the IACUC at Duke University, University of Notre Dame and Princeton University and adhered to all the laws and guidelines of Kenya. For a complete set of acknowledgements of logistical assistance and data collection and management, please visit http://amboselibaboons.nd.edu/acknowledgements/. We also thank the editor and two anonymous reviewers for exceptionally helpfully comments that substantially improved the quality of this article.

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
