## [Reviewer comments · Proceedings of the Royal Society B: Biological Sciences]

Review History

RSPB-2020-1013.R0 (Original submission)

Review form: Reviewer 1

Recommendation

Accept with minor revision (please list in comments)

Scientific importance: Is the manuscript an original and important contribution to its field?

Excellent

General interest: Is the paper of sufficient general interest?

Excellent

Quality of the paper: Is the overall quality of the paper suitable?

Excellent

Is the length of the paper justified?

Yes

Should the paper be seen by a specialist statistical reviewer?

No

Do you have any concerns about statistical analyses in this paper? If so, please specify them explicitly in your report.

No

It is a condition of publication that authors make their supporting data, code and materials available - either as supplementary material or hosted in an external repository. Please rate, if applicable, the supporting data on the following criteria.

Is it accessible?

Yes

Is it clear?

Yes

Is it adequate?

Yes

Do you have any ethical concerns with this paper?

No

Comments to the Author

Dominance hierarchies are a common feature of animal groups, and dominance rank is clearly linked to fitness outcomes in a variety of taxa. Thus, it is important to think about how dominance rank should be quantified, and a number of different methods have been proposed over the years. Generally, decisions about which system to use is made on an ad hoc basis and rely on researchers' intuition about what matters to the animals. This paper takes a very welcome approach to this problem by grounding methods for measuring rank in a theoretical model about the nature of competitive pressures animals encounter.

They consider two popular methods for assessing rank. "Ordinal rank" methods assign animals integer ranks from 1 (top) to n (where n = number of animals in group or age-sex class).

"Proportional rank" measures assign fractional ranks based on the proportion of animals dominated. These measures may mean different things to animals group size varies and competitive conditions vary. Having a proportional rank of 0.5 in a group of 3 may mean something quite different than having the same proportional rank in a group of 15. That is because in the group of 3, the middle rank animal is subordinate to only 1 animal, while in a group of 15, the middle ranking animal is subordinate to 6 individuals (if I have my math right). The authors of this paper propose that ordinal rank will be more relevant to fitness outcomes when competition is density dependent and proportional rank will be more relevant to fitness outcomes when competition is density-independent. If this reasoning is correct, the authors argue, then we will actually be able to reverse engineer inferences about the competitive regime by examining the nature of the relationship between different measures of rank and traits of interest (i.e. access to food or mates). This would be very useful for understanding selective forces influencing animals.

To test their ideas, the authors used long-term data from studies of yellow baboons in Amboseli to compare correlations between 20 traits (ranging from proportion of consortships obtained to wound healing and hormone levels) and ordinal/proportion ranks. They find, overall, that ordinal ranks are more closely correlated with the great majority of these traits. This is a very interesting finding.

This is a very interesting set of findings. If the authors' reasoning is correct, then baboons are much more strongly affected by density-dependent competition over food and mates than density independent competition. This is plausible. However, the paper (and the theoretical model that the authors outlined) would be much more compelling (and more broadly relevant) if the authors had made a priori predictions about which traits would be more affected by density-

dependent competition and which traits would be more affected by density-dependent competition and then compared the fit for the two ranking procedures. This would strengthen the broad goal of being able to use information about correlations with rank measures to infer the nature of the competitive regime.

The authors limit the analysis to ordinal measures. In both of the measures that the authors compare, are based on an ordinal ranking system in which animals are ordered along a continuum (from 1 to n, or from 1.0 to 0) and the distances between adjacent ranks are identical. That is the distance between rank 1 and rank 2 is the same as the distance between rank 7 and rank 8; and the difference between rank 1 and 0.9 is the same as the difference between 0.3 and 0.2. In cardinal systems, like Elo-ranking methods which are now popular, the distances between adjacent individuals are not fixed. You can not only model the order of individuals in the hierarchy, you can assess the whether the distance in rank between individuals varies across the hierarchy. So, in despotic system, the top ranking male might be MUCH more powerful than anyone else, and the subordinates may be more closely clumped together. Given that a lot of people have been using these kinds of methods, I was curious about how cardinal measures would do against the ordinal measures. The authors might have had reasons for limiting their analysis to ordinal measures, but I think they should explain why they have made this decision.

Point of terminology: From my point of view both the methods are ordinal measures. I think it would be better to find another term for what the authors refer to as "ordinal rank" measure (perhaps "integer rank"?)

Review form: Reviewer 2

Recommendation

Major revision is needed (please make suggestions in comments)

Scientific importance: Is the manuscript an original and important contribution to its field?

Acceptable

General interest: Is the paper of sufficient general interest?

Good

Quality of the paper: Is the overall quality of the paper suitable?

Good

Is the length of the paper justified?

Yes

Should the paper be seen by a specialist statistical reviewer?

No

Do you have any concerns about statistical analyses in this paper? If so, please specify them explicitly in your report.

Yes

It is a condition of publication that authors make their supporting data, code and materials available - either as supplementary material or hosted in an external repository. Please rate, if applicable, the supporting data on the following criteria.

Is it accessible?

N/A

Is it clear?

N/A

Is it adequate?

N/A

Do you have any ethical concerns with this paper?

No

Comments to the Author

Main comments:

This study interrogates the underlying assumptions of different dominance metrics, and highlights how acknowledgement of these assumptions can provide insight into the competitive landscapes shaping traits. This is an important endeavour and the assumptions identified are sensible and thoughtful. With identification of the assumptions as the first goal, the second goal was to “identify which rank metric best predicts a wide range of rank-related traits in wild baboons, with the aim of identifying broad patterns of the role of competition in this population”. An impressive dataset was combined to meet this goal. However it is currently unclear to me how the findings meet the aim of identifying broad patterns of the role of competition in this population, beyond stating that the traits are shaped either by density-dependent or density-independent resource access. A table of the dependent variables is provided, but I am missing an interpretation of what these variables and their preferred model (i.e., ordinal or proportional) mean for understanding competitive landscapes, either in the table or in the discussion. Without empirically testing the assumptions laid out in the introduction, it seems necessary to do more than simply state that the traits are shaped by density-dependent or density-independent resources access. For instance, I would like to see some kind of discussion of what resource access each of the traits could be shaped by, or, as done to some extent in lines 392-395, whether the preferred models for the traits are consistent with your assumptions. This study has some interesting findings, particularly that preferred models (i.e., ordinal vs. proportional) differ in males vs. females. I would be very interested in some more interpretation of this in the discussion, and how this could provide insight into different competitive regimes between the sexes. It seems the advantage this study has over those highlighted in lines 443-457 is the opportunity to interpret the findings in light of the assumptions identified in the introduction. Without enough of this, the study appears to provide more of an incremental contribution. Some relevant background information of the baboons social and ecological background could be helpful for doing this (e.g., details on patterns of male/female migration, food availability) as not all readers will be familiar with these details. Overall I think it is necessary to provide a deeper interpretation of the findings for the reader in terms of how they provide insight into competitive regimes.

Other comments:

- In the introduction, you provide useful theoretical scenarios which nicely illustrate the importance in differentiating between ordinal vs. proportional rank. There are some opportunities for improving clarity here. On lines 202-207 you provide a clear example of new males joining a group, making proportional and ordinal ranks no longer interchangeable. I think you are continuing this same example (new males joining a group) in lines 235-238 to illustrate density-dependent resource accessibility, but in lines 270-272 you are now referring to two separate groups with 9 and 5 individuals, not increasing the number of individuals in a single group. It could be clarified where they are separate examples, or continuing with the same running example.
- I am unsure why on line 237 the number of estrus females “increases more slowly than the number of males does...as male hierarchy increases”; if it is just male hierarchy size increasing, the number of females are not increasing at all, but remaining constant.

- It could be beneficial to provide a brief definition or clearer explanation of 'density-dependent/independent', as it is such a key phrase.
- I think only one of the two panels of figure 2 may be necessary as they are both representing essentially the same thing. Also, as the figure caption is long, it would be beneficial to label the black dots on the graph as empirical data from Amboseli.
- Line 333: "We replicated as closely as possible the models presented in the original analysis". Were there different covariates/random effects used across models, and could these differences affect interpretation of the models? It could be useful to provide these models and their parameters in the supplementary material.
- 443-457; it seems these studies should be in the introduction, highlighting how the current study differentiates from these.
- On line 456-457, it states that no previous study has used empirical data to compare how rank metrics perform in predicting traits. However, a study by Funkhouser et al., (2018) also looked at a range of different approaches to measuring the hierarchy (e.g., elo-ratings, ADAGIO) in one group of chimpanzees and Tibetan macaques, and tested how rank predicted social behaviour (affiliation, agonism, nearest neighbour). This study should be cited, with an explanation of how the current study provides novel insights that Funkhouser et al. (2018) does not.
 - o Funkhouser, J. A., Mayhew, J. A., Sheeran, L. K., Mulcahy, J. B., & Li, J. H. (2018). Comparative investigations of social context-dependent dominance in captive chimpanzees (*Pan troglodytes*) and wild Tibetan macaques (*Macaca thibetana*). *Scientific reports*, 8(1), 1-15.

Decision letter (RSPB-2020-1013.R0)

08-Jun-2020

Dear Ms Levy:

Your manuscript has now been peer reviewed and the reviews have been assessed by an Associate Editor. The reviewers' comments (not including confidential comments to the Editor) and the comments from the Associate Editor are included at the end of this email for your reference. The reviewers, AE and I all appreciate the goals of your manuscript, however there are several concerns raised by the reviewers that need to be addressed. In particular, I highlight two of these. First, I agree with reviewer 1 that you need to at least discuss why you chose not to include cardinal rank systems (I also agree with this reviewer that both of your ranks are ordinal scales). Second, both reviewers highlight that you could do quite a bit more with your predictions and interpretation. Of course, both reviewers make other suggestions in their comments, which are appended below, and please ensure that you fully address all of their concerns.

We do not allow multiple rounds of revision so we urge you to make every effort to fully address all of the comments at this stage. We expect to send your manuscript back to one or more of the original reviewers for assessment. If the original reviewers are not available we may invite new reviewers. Please note that this does not guarantee eventual acceptance of your manuscript.

Research ethics:

Use of animals and field studies:

Please submit a copy of your revised paper within three weeks. If we do not hear from you within this time your manuscript will be rejected. If you are unable to meet this deadline please let us know as soon as possible, as we may be able to grant a short extension.

Best wishes,
Dr Sarah Brosnan
Editor, Proceedings B
mailto:proceedingsb@royalsociety.org

Associate Editor
Board Member: 1
Comments to Author:

This paper presents an interesting analysis of dominance rank and how different methods of calculation have different implications for the relative importance of dominance rank in fitness. I think this is novel and of broad importance, but I agree with both reviewers that the novelty and potential application needs to be made clearer to reach the broad audience of this journal. Without careful reframing it is possible that the paper will seem largely incremental and have less impact than it should.

Reviewer(s)' Comments to Author:
Referee: 1

Comments to the Author(s)

Dominance hierarchies are a common feature of animal groups, and dominance rank is clearly linked to fitness outcomes in a variety of taxa. Thus, it is important to think about how dominance rank should be quantified, and a number of different methods have been proposed over the years. Generally, decisions about which system to use is made on an ad hoc basis and rely on researchers' intuition about what matters to the animals. This paper takes a very welcome approach to this problem by grounding methods for measuring rank in a theoretical model about the nature of competitive pressures animals encounter.

They consider two popular methods for assessing rank. "Ordinal rank" methods assign animals integer ranks from 1 (top) to n (where n = number of animals in group or age-sex class). "Proportional rank" measures assign fractional ranks based on the proportion of animals dominated. These measures may mean different things to animals group size varies and competitive conditions vary. Having a proportional rank of 0.5 in a group of 3 may mean something quite different than having the same proportional rank in a group of 15. That is because in the group of 3, the middle rank animal is subordinate to only 1 animal, while in a group of 15, the middle ranking animal is subordinate to 6 individuals (if I have my math right). The authors of this paper propose that ordinal rank will be more relevant to fitness outcomes when competition is density dependent and proportional rank will be more relevant to fitness outcomes when competition is density-independent. If this reasoning is correct, the authors argue, then we will actually be able to reverse engineer inferences about the competitive regime by examining the nature of the relationship between different measures of rank and traits of interest (i.e. access to food or mates). This would be very useful for understanding selective forces influencing animals.

To test their ideas, the authors used long-term data from studies of yellow baboons in Amboseli to compare correlations between 20 traits (ranging from proportion of consortships obtained to wound healing and hormone levels) and ordinal/proportion ranks. They find, overall, that ordinal ranks are more closely correlated with the great majority of these traits. This is a very interesting finding.

This is a very interesting set of findings. If the authors' reasoning is correct, then baboons are much more strongly affected by density-dependent competition over food and mates than density independent competition. This is plausible. However, the paper (and the theoretical model that the authors outlined) would be much more compelling (and more broadly relevant) if the authors had made a priori predictions about which traits would be more affected by density-dependent competition and which traits would be more affected by density-independent competition and then compared the fit for the two ranking procedures. This would strengthen the broad goal of being able to use information about correlations with rank measures to infer the nature of the competitive regime.

The authors limit the analysis to ordinal measures. In both of the measures that the authors compare, are based on an ordinal ranking system in which animals are ordered along a continuum (from 1 to n, or from 1.0 to 0) and the distances between adjacent ranks are identical. That is the distance between rank 1 and rank 2 is the same as the distance between rank 7 and rank 8; and the difference between rank 1 and 0.9 is the same as the difference between 0.3 and 0.2. In cardinal systems, like Elo-ranking methods which are now popular, the distances between adjacent individuals are not fixed. You can not only model the order of individuals in the hierarchy, you can assess the whether the distance in rank between individuals varies across the hierarchy. So, in despotic system, the top ranking male might be MUCH more powerful than anyone else, and the subordinates may be more closely clumped together. Given that a lot of people have been using these kinds of methods, I was curious about how cardinal measures would do against the ordinal measures. The authors might have had reasons for limiting their analysis to ordinal measures, but I think they should explain why they have made this decision.

Point of terminology: From my point of view both the methods are ordinal measures. I think it would be better to find another term for what the authors refer to as "ordinal rank" measure (perhaps "integer rank"?)

Referee: 2

Comments to the Author(s)

Main comments:

This study interrogates the underlying assumptions of different dominance metrics, and highlights how acknowledgement of these assumptions can provide insight into the competitive landscapes shaping traits. This is an important endeavour and the assumptions identified are sensible and thoughtful. With identification of the assumptions as the first goal, the second goal was to "identify which rank metric best predicts a wide range of rank-related traits in wild baboons, with the aim of identifying broad patterns of the role of competition in this population". An impressive dataset was combined to meet this goal. However it is currently unclear to me how the findings meet the aim of identifying broad patterns of the role of competition in this population, beyond stating that the traits are shaped either by density-dependent or density-independent resource access. A table of the dependent variables is provided, but I am missing an interpretation of what these variables and their preferred model (i.e., ordinal or proportional) mean for understanding competitive landscapes, either in the table or in the discussion. Without empirically testing the assumptions laid out in the introduction, it seems necessary to do more than simply state that the traits are shaped by density-dependent or density-independent resources access. For instance, I would like to see some kind of discussion of what resource access each of the traits could be shaped by, or, as done to some extent in lines 392-395, whether the preferred models for the traits are consistent with your assumptions. This study has some interesting findings, particularly that preferred models (i.e., ordinal vs. proportional) differ in males vs. females. I would be very interested in some more interpretation of this in the discussion, and how this could provide insight into different competitive regimes between the

sexes. It seems the advantage this study has over those highlighted in lines 443-457 is the opportunity to interpret the findings in light of the assumptions identified in the introduction. Without enough of this, the study appears to provide more of an incremental contribution. Some relevant background information of the baboons social and ecological background could be helpful for doing this (e.g., details on patterns of male/female migration, food availability) as not all readers will be familiar with these details. Overall I think it is necessary to provide a deeper interpretation of the findings for the reader in terms of how they provide insight into competitive regimes.

Other comments:

- In the introduction, you provide useful theoretical scenarios which nicely illustrate the importance in differentiating between ordinal vs. proportional rank. There are some opportunities for improving clarity here. On lines 202-207 you provide a clear example of new males joining a group, making proportional and ordinal ranks no longer interchangeable. I think you are continuing this same example (new males joining a group) in lines 235-238 to illustrate density-dependent resource accessibility, but in lines 270-272 you are now referring to two separate groups with 9 and 5 individuals, not increasing the number of individuals in a single group. It could be clarified where they are separate examples, or continuing with the same running example.
- I am unsure why on line 237 the number of estrus females “increases more slowly than the number of males does...as male hierarchy increases”; if it is just male hierarchy size increasing, the number of females are not increasing at all, but remaining constant.
- It could be beneficial to provide a brief definition or clearer explanation of ‘density-dependent/independent’, as it is such a key phrase.
- I think only one of the two panels of figure 2 may be necessary as they are both representing essentially the same thing. Also, as the figure caption is long, it would be beneficial to label the black dots on the graph as empirical data from Amboseli.
- Line 333: “We replicated as closely as possible the models presented in the original analysis”. Were there different covariates/random effects used across models, and could these differences affect interpretation of the models? It could be useful to provide these models and their parameters in the supplementary material.
- 443-457; it seems these studies should be in the introduction, highlighting how the current study differentiates from these.
- On line 456-457, it states that no previous study has used empirical data to compare how rank metrics perform in predicting traits. However, a study by Funkhouser et al., (2018) also looked at a range of different approaches to measuring the hierarchy (e.g., elo-ratings, ADAGIO) in one group of chimpanzees and Tibetan macaques, and tested how rank predicted social behaviour (affiliation, agonism, nearest neighbour). This study should be cited, with an explanation of how the current study provides novel insights that Funkhouser et al. (2018) does not.
 - o Funkhouser, J. A., Mayhew, J. A., Sheeran, L. K., Mulcahy, J. B., & Li, J. H. (2018). Comparative investigations of social context-dependent dominance in captive chimpanzees (*Pan troglodytes*) and wild Tibetan macaques (*Macaca thibetana*). *Scientific reports*, 8(1), 1-15.

Author's Response to Decision Letter for (RSPB-2020-1013.R0)

See Appendix A.

RSPB-2020-1013.R1 (Revision)

Review form: Reviewer 2

Recommendation

Accept with minor revision (please list in comments)

Scientific importance: Is the manuscript an original and important contribution to its field?

Excellent

General interest: Is the paper of sufficient general interest?

Excellent

Quality of the paper: Is the overall quality of the paper suitable?

Good

Is the length of the paper justified?

No

Should the paper be seen by a specialist statistical reviewer?

No

Do you have any concerns about statistical analyses in this paper? If so, please specify them explicitly in your report.

No

It is a condition of publication that authors make their supporting data, code and materials available - either as supplementary material or hosted in an external repository. Please rate, if applicable, the supporting data on the following criteria.

Is it accessible?

Yes

Is it clear?

Yes

Is it adequate?

N/A

Do you have any ethical concerns with this paper?

No

Comments to the Author

The current manuscript is much improved and provides a clearer and more stream-lined story, which highlights the value and implications of your findings more clearly. Overall, a much more convincing case has been made in the current manuscript that the findings represent more than an incremental contribution. The points that are of widespread interest have been emphasized more clearly.

- The changes made to improve clarity to the reader make the manuscript flow much better than the previous version.
- Importantly, this version provide a much stronger case that simple ordinal vs. proportional rank predict traits that are shaped by density-dependent and density-independent resources, respectively. The approach of providing the general predictions of in the introduction

(which were implicit in your original version, but helpfully explicit in the current version), while interpreting the specific variables in the discussion, was very effective and highlighted the value of the study.

- The importance and implications of the study are also made clearer by emphasizing that this has generated new hypotheses, and can be used to re-interpret previous research.
- The interpretations of the findings (e.g. in relation to sex-differences) provided in the discussion do the study greater justice than the previous manuscript version by emphasizing the features that are of widespread interest.
- Table S5 is useful for readers interested in the specific traits included your analyses.

A few places where there is opportunity for potential improvement:

- The main reason for predicting that female traits would be predicted by density-independent competition is based on socioecological models. There is a lot of debate regarding the utility of socioecological models. While of course there isn't space to go into this debate here, I think it's worth acknowledging when making this prediction, and indeed, could help with interpreting the fact that only some of the findings were in line with this prediction.
- In the results section it is mentioned that for 35% of traits, a relationship between rank and a trait of interest would have been undetected if researchers had chosen the alternative rank method. This is likely of interest to many readers and could be worth bringing out again in the discussion.
- In line 272 you say that the three female traits better predicted by proportional rank are likely associated with food competition, but it's not immediately clear why. If the cited papers provide evidence for that I would make that clearer.
- The two paragraphs on lines 325-348 could be condensed, or even potentially cut if aiming to shorten the paper.

Decision letter (RSPB-2020-1013.R1)

06-Aug-2020

Dear Ms Levy

I am pleased to inform you that your manuscript RSPB-2020-1013.R1 entitled "A comparison of dominance rank metrics reveals multiple competitive landscapes in an animal society" has been provisionally accepted for publication in Proceedings B, however the reviewers also recommend some minor revisions to your manuscript. Therefore, I invite you to respond to the referee(s)' comments and revise your manuscript. Because the schedule for publication is very tight, it is a condition of publication that you submit the revised version of your manuscript within 7 days. If you do not think you will be able to meet this date please let us know.

Sincerely,
 Dr Sarah Brosnan
 Editor, Proceedings B
<mailto:proceedingsb@royalsociety.org>

Associate Editor:
 Board Member: 1
 Comments to Author:

This paper provides an interesting and novel perspective on dominance rank metrics and will provide a valuable contribution to the literature. R2 has provided some minor suggestions that should be followed to increase the clarity of the paper before publication.

Reviewer(s)' Comments to Author:

Referee: 2
 Comments to the Author(s)

The current manuscript is much improved and provides a clearer and more stream-lined story, which highlights the value and implications of your findings more clearly. Overall, a much more convincing case has been made in the current manuscript that the findings represent more than an incremental contribution. The points that are of widespread interest have been emphasized more clearly.

- The changes made to improve clarity to the reader make the manuscript flow much better than the previous version.
- Importantly, this version provide a much stronger case that simple ordinal vs. proportional rank predict traits that are shaped by density-dependent and density-independent resources, respectively. The approach of providing the general predictions of in the introduction (which were implicit in your original version, but helpfully explicit in the current version), while interpreting the specific variables in the discussion, was very effective and highlighted the value of the study.
- The importance and implications of the study are also made clearer by emphasizing that this has generated new hypotheses, and can be used to re-interpret previous research.
- The interpretations of the findings (e.g. in relation to sex-differences) provided in the discussion do the study greater justice than the previous manuscript version by emphasizing the features that are of widespread interest.
- Table S5 is useful for readers interested in the specific traits included your analyses.

A few places where there is opportunity for potential improvement:

- The main reason for predicting that female traits would be predicted by density-independent competition is based on socioecological models. There is a lot of debate regarding the utility of socioecological models. While of course there isn't space to go into this debate here, I think it's worth acknowledging when making this prediction, and indeed, could help with interpreting the fact that only some of the findings were in line with this prediction.
- In the results section it is mentioned that for 35% of traits, a relationship between rank and a trait of interest would have been undetected if researchers had chosen the alternative rank method. This is likely of interest to many readers and could be worth bringing out again in the discussion.
- In line 272 you say that the three female traits better predicted by proportional rank are likely associated with food competition, but it's not immediately clear why. If the cited papers provide evidence for that I would make that clearer.
- The two paragraphs on lines 325-348 could be condensed, or even potentially cut if aiming to shorten the paper.

Author's Response to Decision Letter for (RSPB-2020-1013.R1)

See Appendix B.

Decision letter (RSPB-2020-1013.R2)

18-Aug-2020

Dear Ms Levy

I am pleased to inform you that your manuscript entitled "A comparison of dominance rank metrics reveals multiple competitive landscapes in an animal society" has been accepted for publication in Proceedings B.

Open Access

Paper charges

Sincerely,
Proceedings B
<mailto:proceedingsb@royalsociety.org>

Appendix A

Dear Dr. Brosnan,

We thank you and the two reviewers for providing comments on our recent submission. We were pleased that you, the Associate Editor, and the reviewers saw merit in our goals. We have revised the manuscript in response to your comments, and those of the Associate Editor and reviewers, and we believe that the paper is much improved. Our responses to each concern are provided in detail below, with point-by-point responses in indented text following each original comment. In the manuscript, we have highlighted added or edited text in grey.

Reviews

Your manuscript has now been peer reviewed and the reviews have been assessed by an Associate Editor. The reviewers' comments (not including confidential comments to the Editor) and the comments from the Associate Editor are included at the end of this email for your reference. The reviewers, AE and I all appreciate the goals of your manuscript, however there are several concerns raised by the reviewers that need to be addressed. In particular, I highlight two of these. First, I agree with reviewer 1 that you need to at least discuss why you chose not to include cardinal rank systems (I also agree with this reviewer that both of your ranks are ordinal scales). Second, both reviewers highlight that you could do quite a bit more with your predictions and interpretation. Of course, both reviewers make other suggestions in their comments, which are appended below, and please ensure that you fully address all of their concerns.

Response: Thank you for this summary. We have responded to all comments below, with specific attention to the two points you highlight here: discussing our decision to not include cardinal ranks, and expanding our discussion of our predictions and interpretations.

Associate Editor

Board Member: 1

Comments to Author:

This paper presents an interesting analysis of dominance rank and how different methods of calculation have different implications for the relative importance of dominance rank in fitness. I think this is novel and of broad importance, but I agree with both reviewers that the novelty and potential application needs to be made clearer to reach the broad audience of this journal. Without careful reframe it is possible that the paper will seem largely incremental and have less impact than it should.

Response: Thank you for your suggestions. We have taken several measures to highlight the novelty and application of this paper, as you and both reviewers suggested.

In the Introduction, we edited the text introducing dominance ranks (lines 26-30), further clarified our framework (lines 110-113), explicitly stated our predictions (lines 119-126), highlighted the novelty of the study (lines 127-132), and explained how this study and method increase our understanding of competition (lines 132-134).

In the Discussion, we interpreted our findings in the context of our predictions (lines 254-278), further described the utility of this method (lines 284-289), and suggest a comparable framework for cardinal rank metrics as well as room for future study (lines 348-358).

Reviewer(s)' Comments to Author:

Referee: 1

Comments to the Author(s)

Dominance hierarchies are a common feature of animal groups, and dominance rank is clearly linked to fitness outcomes in a variety of taxa. Thus, it is important to think about how dominance rank should be quantified, and a number of different methods have been proposed over the years. Generally, decisions about which system to use is made on an ad hoc basis and rely on researchers' intuition about what matters to the animals. This paper takes a very welcome approach to this problem by grounding methods for measuring rank in a theoretical model about the nature of competitive pressures animals encounter.

They consider two popular methods for assessing rank. "Ordinal rank" methods assign animals integer ranks from 1 (top) to n (where n = number of animals in group or age-sex class). "Proportional rank" measures assign fractional ranks based on the proportion of animals dominated. These measures may mean different things to animals group size varies and competitive conditions vary. Having a proportional rank of 0.5 in a group of 3 may mean something quite different than having the same proportional rank in a group of 15. That is because in the group of 3, the middle rank animal is subordinate to only 1 animal, while in a group of 15, the middle ranking animal is subordinate to 6 individuals (if I have my math right).

The authors of this paper propose that ordinal rank will be more relevant to fitness outcomes when competition is density dependent and proportional rank will be more relevant to fitness outcomes when competition is density-independent. If this reasoning is correct, the authors argue, then we will actually be able to reverse engineer inferences about the competitive regime by examining the nature of the relationship between different measures of rank and traits of interest (i.e. access to food or mates). This would be very useful for understanding selective forces influencing animals.

To test their ideas, the authors used long-term data from studies of yellow baboons in Amboseli to compare correlations between 20 traits (ranging from proportion of consortships obtained to wound healing and hormone levels) and ordinal/proportion ranks. They find, overall, that ordinal ranks are more closely correlated with the great majority of these traits. This is a very interesting finding.

Response: Thank you for your positive summary and interest in the project. We appreciate it.

This is a very interesting set of findings. If the authors' reasoning is correct, then baboons are much more strongly affected by density-dependent competition over food and mates than density independent competition. This is plausible. However, the paper (and the theoretical model that the authors outlined) would be much more compelling (and more broadly relevant) if the authors had made a priori predictions about which traits would be more affected by density-dependent competition and which traits would be more affected by density-independent competition and then compared the fit for the two ranking procedures. This would strengthen the broad goal of being able to use information about correlations with rank measures to infer the nature of the competitive regime.

Response: To address this comment, we added two specific predictions to the Introduction (lines 119-126). We were careful not to make post-hoc predictions about each trait, as we did

not do so at the outset of this project. Instead, we highlight two predictions that underlie our motivations for this study but were not explicit in the original version.

First, we predicted that male mate competition would be best predicted by ordinal rank. This prediction was based on: (i) the observation that estrus females do not increase in abundance in proportion with male group size (Figure 2); (ii) prior discoveries, from our and other baboon populations, that ordinal dominance rank is a robust predictor of male mating success; and (iii) the observation that estrus female baboons can be monopolized by a single adult male.

Second, we expected that that female traits would be better predicted by proportional rank as compared to ordinal rank. This prediction was based on: (i) socioecological models that predict that food resources are the most salient competitive arena for females (Emlen & Oring, 1977); and (ii) the observation that, as a primate group increases in size, the resource base (food) available to that group also increases (Isbell, 1991).

We have added two paragraphs to the Discussion that explain how our predictions are supported by results of specific traits (lines 254-278). We also highlight that our results generate new hypotheses about the specific ways that each trait is related to within-group competition. For example, we generate a hypothesis about competition for social partners in lines 284-289.

The authors limit the analysis to ordinal measures. In both of the measures that the authors compare, are based on an ordinal ranking system in which animals are ordered along a continuum (from 1 to n, or from 1.0 to 0) and the distances between adjacent ranks are identical. That is the distance between rank 1 and rank 2 is the same as the distance between rank 7 and rank 8; and the difference between rank 1 and 0.9 is the same as the difference between 0.3 and 0.2. In cardinal systems, like Elo-ranking methods which are now popular, the distances between adjacent individuals are not fixed. You can not only model the order of individuals in the hierarchy, you can assess the whether the distance in rank between individuals varies across the hierarchy. So, in despotic system, the top ranking male might be MUCH more powerful than anyone else, and the subordinates may be more closely clumped together. Given that a lot of people have been using these kinds of methods, I was curious about how cardinal measures would do against the ordinal measures. The authors might have had reasons for limiting their analysis to ordinal measures, but I think they should explain why they have made this decision.

Response: We agree that cardinal rank metrics deserve a greater discussion than the passing mention that we had previously given them in the Introduction. We now include more information about cardinal ranks in the Introduction (lines 31-33), and we note that a full treatment of cardinal ranks is beyond the scope of this paper (lines 44-46).

We have also added a paragraph to the Discussion on lines 348-358 regarding potential extensions of our framework to cardinal rank metrics, including a discussion of absolute Elo scores versus Elo scores that have been normalized based on group size or within-group variation in Elo scores. We highlight this extension as a promising avenue of future research.

Point of terminology: From my point of view both the methods are ordinal measures. I think it would be better to find another term for what the authors refer to as “ordinal rank” measure (perhaps “integer rank”?)

Response: We agree with the reviewer that both “ordinal rank” and “proportional rank” refer to ordered rank measures. We have re-written a paragraph in the Introduction (lines 26-30) to emphasize that both of the measures that we refer to are measures derived from an animal’s order in a dominance hierarchy. In addition, we have changed how we refer to the metric we originally called “ordinal rank”; it is now called “simple ordinal rank” reflecting that this rank metric is just a simple integer ordering of animals in a hierarchy. We carry this change throughout the manuscript wherever we refer to the simple ordinal rank metric.

Referee: 2

Comments to the Author(s)

Main comments:

This study interrogates the underlying assumptions of different dominance metrics, and highlights how acknowledgement of these assumptions can provide insight into the competitive landscapes shaping traits. This is an important endeavour and the assumptions identified are sensible and thoughtful. With identification of the assumptions as the first goal, the second goal was to “identify which rank metric best predicts a wide range of rank-related traits in wild baboons, with the aim of identifying broad patterns of the role of competition in this population”. An impressive dataset was combined to meet this goal.

However it is currently unclear to me how the findings meet the aim of identifying broad patterns of the role of competition in this population, beyond stating that the traits are shaped either by density-dependent or density-independent resource access.

Response: We agree that identifying density-dependent versus density-independent patterns of resource distribution and access is only one component of understanding competition in complex societies. We have modified the wording of our second goal to say “Second, we identify which rank metric (simple ordinal or proportional) best predicts a wide range of rank-related traits in wild baboons, with the aim of better understanding density-dependent versus density-independent patterns of resource distribution and access in a complex animal society.” (lines 53-55). In addition, to better explain how our study helps to identify these patterns in density-independent and density-dependent competition, we have added text in the Discussion to more explicitly examine the implications of our results for understanding broad patterns of competition (lines 254-278).

A table of the dependent variables is provided, but I am missing an interpretation of what these variables and their preferred model (i.e., ordinal or proportional) mean for understanding competitive landscapes, either in the table or in the discussion. Without empirically testing the assumptions laid out in the introduction, it seems necessary to do more than simply state that the traits are shaped by density-dependent or density-independent resources access. For instance, I would like to see some kind of discussion of what resource access each of the traits could be shaped by, or, as done to some extent in lines 392-395, whether the preferred models for the traits are consistent with your assumptions.

Response: We appreciate this comment and we feel that addressing it strengthened the manuscript. We have added substantial text to the Discussion that both interprets our results in the context of our predictions, and also interprets some of the specific findings (mentioned above; lines 254-278). In doing so, we highlight the fact that our interpretations of specific traits

are also new hypotheses that can be tested. This means that for any trait, our findings (i) help us understand the mode of competition acting to produce that trait (density-dependent or independent), and (ii) allow us to generate new hypotheses that can be tested to understand the key factors and resources (e.g., food access) that lead to one rank metric being preferred over another and, hence, drive one mode of competition relative to the other.

This study has some interesting findings, particularly that preferred models (i.e., ordinal vs. proportional) differ in males vs. females. I would be very interested in some more interpretation of this in the discussion, and how this could provide insight into different competitive regimes between the sexes.

Response: We have added substantial text to the Discussion to enhance the interpretation of our results in the context of our predictions and interpret specific findings (mentioned above; lines 254-278).

It seems the advantage this study has over those highlighted in lines 443-457 is the opportunity to interpret the findings in light of the assumptions identified in the introduction. Without enough of this, the study appears to provide more of an incremental contribution. Some relevant background information of the baboons social and ecological background could be helpful for doing this (e.g., details on patterns of male/female migration, food availability) as not all readers will be familiar with these details. Overall I think it is necessary to provide a deeper interpretation of the findings for the reader in terms of how they provide insight into competitive regimes.

Response: In the new paragraphs that interpret our findings (mentioned above; lines 254-278), we include examples and hypotheses that put our findings in the context of baboon life history. We hope that these additions, in conjunction with the description of male competition for estrous females in the Introduction, and the first paragraph of the Methods section address your suggestion.

Other comments:

- In the introduction, you provide useful theoretical scenarios which nicely illustrate the importance in differentiating between ordinal vs. proportional rank. There are some opportunities for improving clarity here. On lines 202-207 you provide a clear example of new males joining a group, making proportional and ordinal ranks no longer interchangeable. I think you are continuing this same example (new males joining a group) in lines 235-238 to illustrate density-dependent resource accessibility, but in lines 270-272 you are now referring to two separate groups with 9 and 5 individuals, not increasing the number of individuals in a single group. It could be clarified where they are separate examples, or continuing with the same running example.

Response: We have clarified the text in two places to help with continuity of the original example. In the paragraph describing ordinal rank, we added the sentence “In other words, the 5th-ranking male in a hierarchy of 5 will have the same mating access as the 5th-ranking male in a hierarchy of 9” (lines 93-95). In the paragraph describing proportional rank, we changed the wording to read: “This situation might occur, for instance, in competition for food if a hierarchy grows from 5 to 9 individuals and its home range nearly doubles in size (with nearly twice the amount of food). Unlike our mate competition example, the 3rd-ranking individual in the hierarchy of 5 has approximately equal access to food as the 5th-ranking individual in the group of 9” (lines 101-104).

- I am unsure why on line 237 the number of estrus females “increases more slowly than the number of males does...as male hierarchy increases”; if it is just male hierarchy size increasing, the number of females are not increasing at all, but remaining constant.

Response: We agree that our description was not clear. We have revised the text to say, “As group size increases, the daily availability of estrous females increases more slowly than either the number of adult females or the number of adult males in the group” (lines 87-89).

- It could be beneficial to provide a brief definition or clearer explanation of ‘density-dependent/independent’, as it is such a key phrase.

Response: These are critically important terms and they are somewhat challenging to keep straight. We have added a sentence in the Introduction that explicitly defines each: “Density-dependent competition occurs when average per-capita resource access depends on hierarchy size (i.e., when resources do not increase proportionately with increases in hierarchy size), while density-independent competition occurs when average per-capita resource access is independent of hierarchy size (because resources increase proportionately with hierarchy size).” (lines 110-113).

- I think only one of the two panels of figure 2 may be necessary as they are both representing essentially the same thing. Also, as the figure caption is long, it would be beneficial to label the black dots on the graph as empirical data from Amboseli.

Response: We agree that the two panels in figure 2 are somewhat redundant, and the combination makes the figure caption quite long. However, in describing this framework in talks and to colleagues, we found that some people were able to understand the concept better when thinking about total resource base, while others preferred to think about per capita resource access (panels A and B, respectively). We have therefore retained both panels, and we have added a small key for the black dots, as suggested, which helps make the figure more interpretable on its own.

- Line 333: “We replicated as closely as possible the models presented in the original analysis”. Were there different covariates/random effects used across models, and could these differences affect interpretation of the models? It could be useful to provide these models and their parameters in the supplementary material.

Response: Thank you for this suggestion. We have created Table S5 which describes the equations used to model each trait. We agree that this will help readers understand our study more fully.

- 443-457; it seems these studies should be in the introduction, highlighting how the current study differentiates from these.

Response: We have added text about prior studies of dominance rank metrics (which do not predict traits but rather serve as more methodological studies) to the Introduction: “To date, only a handful of studies have tested the ability of different rank metrics to predict traits of interest ([50,56,57,64–66]). Several other studies have used simulated or empirical data to

assess whether different rank metrics produce different hierarchies, given the same data (e.g. [67–71])” (lines 127-19).

- On line 456-457, it states that no previous study has used empirical data to compare how rank metrics perform in predicting traits. However, a study by Funkhouser et al., (2018) also looked at a range of different approaches to measuring the hierarchy (e.g., elo-ratings, ADAGIO) in one group of chimpanzees and Tibetan macaques, and tested how rank predicted social behaviour (affiliation, agonism, nearest neighbour). This study should be cited, with an explanation of how the current study provides novel insights that Funkhouser et al. (2018) does not.

o Funkhouser, J. A., Mayhew, J. A., Sheeran, L. K., Mulcahy, J. B., & Li, J. H. (2018). Comparative investigations of social context-dependent dominance in captive chimpanzees (*Pan troglodytes*) and wild Tibetan macaques (*Macaca thibetana*). *Scientific reports*, 8(1), 1-15.

Response: Thank you for suggesting Funkhouser et al. (2018); we were not aware of this paper. We now cite it in the Introduction (lines 127-128) and the Discussion (line 323). Similar our study, it runs regressions of the ability of several different ‘dominance contexts’ to predict behavioral data (e.g., ‘all agonism’, ‘affiliation’, ‘nearest neighbor’), and the authors interpret the comparisons that were statistically significant in a behavioral context. Unlike our study, FunkHouser et al. it does not interpret the performance of their rank metrics in the context of density-dependent and independent competition, or in a general theoretical framework.

Appendix B

Dear Dr. Brosnan,

We are glad that you and the reviewers found so much to like in the revised version of our manuscript. The comments from the first round of revision were very helpful in strengthening the manuscript.

We have incorporated Reviewer 2's suggestions from this round of revision into the manuscript and detail our responses to their suggestions below:

Referee: 2

The current manuscript is much improved and provides a clearer and more stream-lined story, which highlights the value and implications of your findings more clearly. Overall, a much more convincing case has been made in the current manuscript that the findings represent more than an incremental contribution. The points that are of widespread interest have been emphasized more clearly.

Response: We thank the reviewer for carefully reading our manuscript and providing such helpful suggestions throughout the review process.

A few places where there is opportunity for potential improvement:

- The main reason for predicting that female traits would be predicted by density-independent competition is based on socioecological models. There is a lot of debate regarding the utility of socioecological models. While of course there isn't space to go into this debate here, I think it's worth acknowledging when making this prediction, and indeed, could help with interpreting the fact that only some of the findings were in line with this prediction.

Response: We agree that it is both (1) important to acknowledge that female-female competition is likely influenced by factors other than food access, and (2) that space prevents an in-depth discussion of these factors.

We have therefore added a statement in the introduction that factors other than food access are involved in female-female competition (lines 124-128). As suggested, we have also added a clause in the discussion that points to these non-food factors as perhaps explaining the results in females (line 283).

- In the results section it is mentioned that for 35% of traits, a relationship between rank and a trait of interest would have been undetected if researchers had chosen the alternative rank method. This is likely of interest to many readers and could be worth bringing out again in the discussion.

Response: We really like this suggestion and have added a sentence that reiterates this result in the first paragraph in the discussion (lines 249-251).

- In line 272 you say that the three female traits better predicted by proportional rank are likely associated with food competition, but it's not immediately clear why. If the cited papers provide evidence for that I would make that clearer.

Response: To help clarify, we have added a parenthetical following each trait, explaining the connection between each trait and food access on lines 276-278.

- The two paragraphs on lines 325-348 could be condensed, or even potentially cut if aiming to shorten the paper.

Response: We agree that there is room here for cutting text. We have highlighted in gray a 7-line section 340-346 that could be cut to save space if necessary, pending the judgment of the editor.

A version of the manuscript with track changes follows below, starting on the next page.